# TOWARDS NOISE-RESISTANT OBJECT DETECTION WITH NOISY ANNOTATIONS

## ABSTRACT

Training deep object detectors requires large amounts of human-annotated images with accurate object labels and bounding box coordinates, which are extremely expensive to acquire. Noisy annotations are much more easily accessible, but they could be detrimental for learning. We address the challenging problem of training object detectors with noisy annotations, where the noise contains a mixture of label noise and bounding box noise. We propose a learning framework which jointly optimizes object labels, bounding box coordinates, and model parameters by performing alternating noise correction and model training. To disentangle label noise and bounding box noise, we propose a two-step noise correction method. The first step performs class-agnostic bounding box correction, and the second step performs label correction and class-specific bounding box refinement. We conduct experiments on PASCAL VOC and MS-COCO dataset with both synthetic noise and machine-generated noise. Our method achieves state-of-the-art performance by effectively cleaning both label noise and bounding box noise [1].

## 1 INTRODUCTION

The remarkable success of modern object detectors largely relies on large-scale datasets with extensive bounding box annotations. However, it is extremely expensive and time-consuming to acquire high-quality human annotations. For example, annotating each bounding box in ILSVRC requires 42s on Mechanical Turk (Su et al., 2012), whereas the recent OpenImagesV4 Kuznetsova et al. (2018) reports 7.4 seconds with extreme clicking (Papadopoulos et al., 2017b). On the other hand, there are ways to acquire annotations at lower costs, such as limiting the annotation time, reducing the number of annotators, or using machine-generated annotations. However, these methods would yield annotations with both *label noise* (*i.e.* wrong classes) and *bounding box noise* (*i.e.* inaccurate locations), which could be detrimental for learning.

Learning with label noise has been an active area of research. Some methods perform label correction using the predictions from the model and modify the loss accordingly (Reed et al., 2015; Tanaka et al., 2018). Other methods treat samples with small loss as those with clean labels, and only allow clean samples to contribute to the loss (Jiang et al., 2018b; Han et al., 2018). However, most of those methods focus on the image classification task where the existence of an object is guaranteed.

Several recent works have studied object detection with noisy annotations. Zhang et al. (2019) focus on the weakly-supervised (WS) setting where only image-level labels are available, and find reliable bounding box instances as those with low classification loss. Gao et al. (2019) study a semi-supervised (SS) setting where the training data contains a small amount of fully-labeled bounding boxes and a large amount of image-level labels, and propose to distill knowledge from a detector pretrained on clean annotations. However, these methods require access to some clean annotations.

In this work, we address a more challenging and practical problem, where the annotation contains *an unknown mixture* of label noise and bounding box noise. Furthermore, *we do not assume access to any clean annotations*. The entanglement of label noise and bounding box noise increases the difficulty to perform noise correction. A commonly used noise indicator, namely the classification loss, is incapable to distinguish label noise from bounding box noise. Furthermore, it is problematic to correct noise directly using the model predictions, because label correction requires accurate

---

[1] Code will be released.

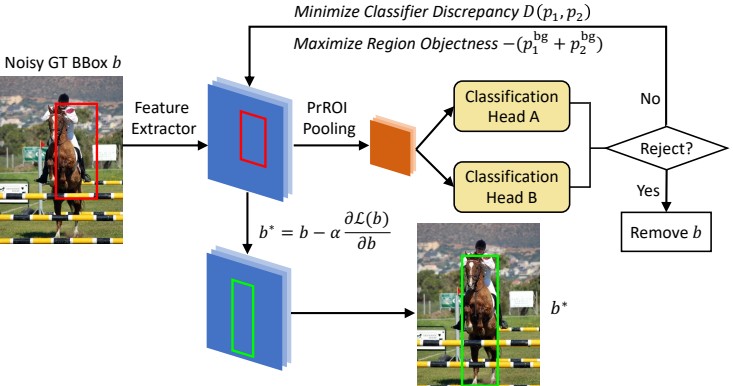

Figure 1: Our Class-Agnostic Bounding Box Correction (CA-BBC) disentangles bounding box (bbox) noise from label noise, by directly optimizing the noisy bbox coordinates regardless of its class label. We use two diverged classifiers to predict the same image region, and update the bbox $b$ to $b^*$ by minimizing classifier discrepancy and maximizing region objectness. Boxes with very low objectness are rejected as false positives.

bounding box coordinates to crop the object, whereas bounding box correction requires accurate class labels to produce the regression offset.

To overcome these difficulties, we propose a two-step noise correction procedure. In the first step, we perform class-agnostic bounding box correction (CA-BBC), which seeks to decouple bounding box noise from label noise, and optimize the noisy ground-truth (GT) bounding box regardless of its class label. An illustration of CA-BBC is shown in Figure 1. It is based on the following intuition: if a bounding box tightly covers an object, then two diverged classifiers would agree with each other and produce the same prediction. Furthermore, both classifiers would have low scores for the background class, *i.e.*, high objectness scores. Therefore, we directly regress the noisy GT bounding box to minimize both classifier discrepancy and background scores. CA-BBC also has the option to reject a bounding box as false positive if the objectness score is too low.

In the second step, we leverage the model's output for label noise correction and class-specific bounding box refinement. It has been shown that co-training two models can filter different types of noise and help each other learn (Blum & Mitchell, 1998; Han et al., 2018; Yu et al., 2019; Chadwick & Newman, 2019). Therefore, we distil knowledge from the ensemble of dual detection heads for noise correction, by generating soft labels and bounding box offsets. We show that soft labels with well-adjusted temperature lead to better performance even for a clean dataset.

To summarize, this paper proposes a noise-resistant learning framework to train object detectors with noisy annotations. The proposed framework jointly optimizes object labels, bounding box coordinates, and model parameters by performing alternating noise correction and model training. We conduct experiments on two benchmarks: PASCAL VOC and MS-COCO, which contain different levels of synthetic noise as well as machine-generated noise. The proposed method outperforms previous methods by a large margin. We also provide qualitative results to demonstrate the efficacy of the two-step noise correction, and ablation studies to examine the effect of each component.

## 2 RELATED WORK

### 2.1 CROWDSOURCING FOR OBJECT DETECTION

Crowdsourcing platforms such as Amazon Mechanical Turk (AMT) have enabled the collection of large-scale datasets. Due to the formidable cost of human annotation, many efforts have been devoted to reduce the annotation cost. However, even an efficient protocol still report 42.4s to annotate one object in an image (Su et al., 2012). Other methods have been proposed which trade off annotation quality for lower cost, by using click supervision (Papadopoulos et al., 2017a), human-in-the-loop labeling (Russakovsky et al., 2015; Papadopoulos et al., 2016; Konyushkova et al., 2018), or exploiting eye-tracking data (Papadopoulos et al., 2014). These methods focus on reducing human effort, rather than combating the annotation noise as our method does.

## 2.2 LEARNING WITH LABEL NOISE

Deep Neural Networks (DNNs) can easily overfit to noisy labels in the training data, leading to poor generalization performance (Zhang et al., 2017). Many works have addressed learning with label noise. Some approaches correct noise by relabeling the noisy samples (Vahdat, 2017; Veit et al., 2017; Lee et al., 2018), but they rely on a small set of clean samples for noise correction. Iterative relabeling methods (Tanaka et al., 2018; Yi & Wu, 2019) have been proposed which produce hard or soft labels using the model predictions. Other approaches filter noise by reweighting or selecting training samples (Jiang et al., 2018b; Ren et al., 2018; Chen et al., 2019b; Arazo et al., 2019; Li et al., 2020). Since DNNs learn clean samples faster than noisy ones, samples with smaller classification loss are usually considered to be clean (Arpit et al., 2017). To avoid error accumulation during the noise correction process, co-teaching (Han et al., 2018) trains two networks simultaneously, where each network selects small-loss samples to train the other. Co-teaching+ (Yu et al., 2019) further keeps the two networks diverged by training on disagreement data.

## 2.3 WEAKLY-SUPERVISED AND SEMI-SUPERVISED OBJECT DETECTION

Weakly-supervised object detection aims to learn object detectors with only image-level labels. Most existing works formulate it as a multiple instance learning (MIL) task (Dietterich et al., 1997), where each label is assigned to a bag of object proposals. A common pipeline is to iteratively alternate between mining object instances using a detector and training the detector using the mined instances (Deselaers et al., 2010; Cinbis et al., 2017). To address the localization noise in the object proposals, Zhang et al. (2019) propose an adaptive sampling method which finds reliable instances as those with high classification scores, and use the reliable instances to impose a similarity loss on noisy images. Different from weakly-supervised object detection which assumes that the correct object label is given, our method deals with label noise and bounding box noise at the same time.

Semi-supervised methods train object detectors using training data with bounding box annotations for some images and only image-level labels for other images (Hoffman et al., 2014; Tang et al., 2016; Uijlings et al., 2018; Gao et al., 2019). Gao et al. (2019) propose an iterative training-mining framework consisting of detector initialization, box mining, and detector retraining. To address the annotation noise of the mined boxes, they use a detector pretrained on clean annotations for knowledge distillation. Different from all semi-supervised learning methods, our method does not need access to any clean annotations.

# 3 METHOD

## 3.1 OVERVIEW

Given a training dataset with images $\mathcal{X}$, noisy object labels $\mathcal{Y}$, and noisy bounding boxes $\mathcal{B}$, our method aims to train an object detector parameterized by $\Theta$, by jointly optimizing $\mathcal{Y}$, $\mathcal{B}$ and $\Theta$. We first warm-up $\Theta$ where we train the detector in a standard manner using the original noisy annotations. After the warm-up, we perform alternating optimization on the annotations and the model. Specifically, for each mini-batch of data $X = \{x_i\}$, $Y = \{y_i\}$, $B = \{b_i\}$, we first keep $\Theta$ fixed and perform noise correction to update $Y$ and $B$, then we used the corrected annotations to update $\Theta$. An overview of the algorithm is shown in Algorithm 1.

We use a popular two-stage object detector (*i.e.* Faster-RCNN (Ren et al., 2015)), which consists of a backbone feature extractor parameterized by $\theta_{\mathrm{cnn}}$, a Region Proposal Network (RPN) $\theta_{\mathrm{rpn}}$, a classification head $\theta_c$, and a bounding box (bbox) regression head $\theta_b$. Note that $\theta_c$ and $\theta_b$ have shared layers. Let *detection head* with parameters $\theta_d$ denote the union of the classification head and the bbox regression head. During training, we simultaneously train two detection heads $\theta_d^1 = \{\theta_c^1, \theta_b^1\}$ and $\theta_d^2 = \{\theta_c^2, \theta_b^2\}$, which are kept diverged from each other by different (random) parameter initializations and different (random) training instance (*i.e.* RoI) sampling.

Due to the entanglement of an unknown mixture of label noise and bbox noise, it is difficult to correct both types of noise in a single step. Therefore, we propose a two-step noise correction method. In the first step, we perform class-agnostic bounding box correction (CA-BBC), which disentangles bbox noise from label noise. In the second step, we utilize the outputs from dual detection heads for label noise correction and class-specific bbox refinement. Figure 2 shows an illustration of our framework. Next we delineate the details.

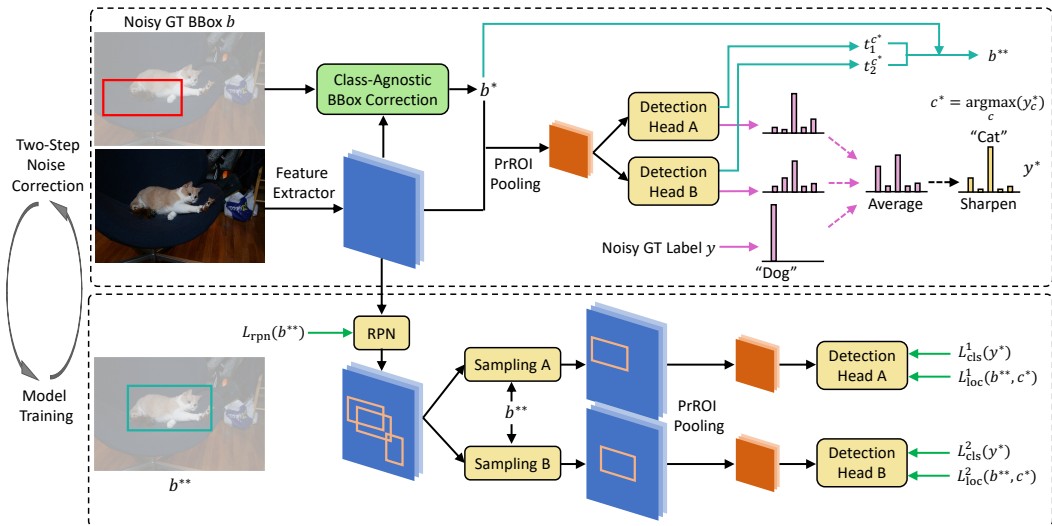

Figure 2: The proposed framework alternately performs noise correction (with fixed model parameters) and model training (with corrected annotations) for each mini-batch. The noise correction procedure consists of two steps: (1) the class-agnostic bounding box correction (Figure 1) disentangles bbox noise and label noise; (2) the class-specific correction step uses dual detection heads to generate soft labels for label correction and refine bounding boxes using class-specific bbox offsets. The two detection heads are kept diverged by different random parameter initialization and different random RoI sampling during training.

---

**Algorithm 1:** alternating two-step noise correction and model training.

1 **Input:** model $\Theta = \{\theta_{\mathrm{cnn}}, \theta_{\mathrm{rpn}}, \theta_d^1, \theta_d^2\}$, noisy training dataset $(\mathcal{X}, \mathcal{Y}, \mathcal{B})$.
2 **while** not MaxIters **do**
3      Mini-batch $X = \{x_i\}, Y = \{y_i\}, B = \{b_i\}$.
4      **for** $b$ **in** $B$ **do**
5          Update $b \to b^*$ with CA-BBC (Eq. 2 & 3).
6      **end**
7      **for** $(y, b^*)$ **in** $(Y, B^*)$ **do**
8          Update $y \to y^*$ with dual-head soft label correction (Eq. 4 & 5).
9          Update $b^* \to b^{**}$ with class-specific bbox refinement (Eq. 6).
10      **end**
11      Update $\Theta$ by SGD on $L_{\mathrm{rpn}}(B^{**})$, $L_{\mathrm{cls}}^{1+2}(Y^*)$, $L_{\mathrm{loc}}^{1+2}(B^{**}, Y^*)$.
12 **end**

---

## 3.2 CLASS-AGNOSTIC BOUNDING BOX CORRECTION

We first correct bounding box noise by updating $B \to B^*$ regardless of the label noise in $Y$. As illustrated in Figure 1, CA-BBC uses two diverged classification heads to produce two sets of class predictions on the same image region, and updates the bounding box to *minimize classifier discrepancy* and *maximize region objectness*. The intuition is: if a bounding box tightly covers an object, then two classifiers would agree with each other and produce the same predictions. Moreover, both predictions would have low scores on the background class.

Specifically, given an image $x \in X$, the backbone first extracts a convolutional feature map. For each noisy GT bounding box $b \in B$, we perform a RoI-Pooling operation on the feature map to extract a fixed-sized feature $\phi(x, b)$. Then we give the RoI feature to the two classification heads to produce two sets of softmax predictions over $C + 1$ classes (including the background class), $p_1(\phi(x, b); \theta_c^1)$ and $p_2(\phi(x, b); \theta_c^2)$. For simplicity we denote them as $p_1$ and $p_2$. The discrepancy between the two predictions is defined as their L2 distance:

$$\mathcal{D}(p_1, p_2) = \|p_1 - p_2\|_2^2. \tag{1}$$

Minimizing the classifier discrepancy *w.r.t* the bounding box will push it to a region where the two classifiers agree on its class label. To prevent the bounding box from simply moving to a background region, we also minimize the classifiers' scores on the background class, $p_1^{\text{bg}}$ and $p_2^{\text{bg}}$. In other words, we want to maximize the objectness of the region covered by the bounding box.

Therefore, we aim to find the optimal $b^*$ that minimizes the following objective function:

$$\mathcal{L}(b) = \mathcal{D}(p_1, p_2) + \lambda(p_1^{\text{bg}} + p_2^{\text{bg}}), \tag{2}$$

where $\lambda$ controls the balance of the two terms and is set to 0.1 in our experiments.

For faster speed, we estimate $b^*$ by performing a single step of gradient descent to update $b$:

$$b^* = b - \alpha \frac{\partial \mathcal{L}(b)}{\partial b}, \tag{3}$$

where $\alpha$ is the step size.

Since RoI-Pooling (Ren et al., 2015) or RoI-Align (He et al., 2017) performs discrete sampling on the feature map to generate $\phi(x, b)$, $\mathcal{L}(b)$ is not differentiable w.r.t $b$. Therefore, we adopt the Precise RoI-Pooling method (Jiang et al., 2018a), which avoids any quantization of coordinates and has a continuous gradient on $b$.

In order to handle false positive bboxes that do not cover any object, we add a reject option which removes $b$ from the ground-truth if both classifiers give a low objectness score (high background score), $p_1^{\text{bg}} > 0.9$ and $p_2^{\text{bg}} > 0.9$.

### 3.3 DUAL-HEAD DISTILLATION FOR NOISE CORRECTION

In the second step, we perform class-specific self-distillation for label noise correction and bbox refinement. We simultaneously train two diverged detection heads which can filter different types of noise, and distil knowledge from their ensemble to clean the annotation noise. Using the ensemble of two heads helps alleviate the confirmation bias problem (*i.e.* a model confirms its own mistakes) that commonly occurs in self-training.

**Soft label correction.** Given the RoI feature $\phi(x, b^*)$, the two classification heads produce two sets of softmax predictions over object classes, $p_1^*$ and $p_2^*$. Inspired by the bootstrapping method (Reed et al., 2015), we use the classifiers' predictions to update the noisy GT label. Let $y \in \{0, 1\}^C$ represent the GT label as a one-hot vector over $C$ classes, we create the soft label by first averaging the classifiers' predictions and the GT label:

$$\bar{y} = (p_1^* + p_2^* + y)/3. \tag{4}$$

Then we apply a sharpening function on the soft label to reduce the entropy of the label distribution. The sharpening operation is defined as:

$$y^* = \bar{y}^{c\frac{1}{T}} \Big/ \sum_{c=1}^{C} \bar{y}^{c\frac{1}{T}}, \ c = 1, 2, ..., C, \tag{5}$$

where $\bar{y}^c$ is the score for class $c$. The temperature $T$ controls the 'softness' of the label and is set to 0.4 in our experiments. A lower temperature decreases the softness and has the implicit effect of entropy minimization, which encourages the model to produce high confidence predictions and allows better decision boundary to be learned (Grandvalet & Bengio, 2005; Berthelot et al., 2019).

**Class-specific bounding box refinement.** The two bbox regression heads produce two sets of per-class bounding box regression offsets, $t_1$ and $t_2$. Let $c^*$ denote the class with the highest score in the soft label, *i.e.* $c^* = \arg\max_c y_c^*, c = 1, 2, ..., C$. We refine the bounding box $b^*$ by merging the class-specific outputs from both bbox regression heads:

$$\begin{aligned} t &= (t_1^{c^*} + t_2^{c^*})/2 \\ b^{**} &= b^* + \rho t, \end{aligned} \tag{6}$$

where $t_1^{c^*}$ and $t_2^{c^*}$ are the bounding box offsets for class $c^*$, and $\rho$ controls the magnitude of the refinement.

| BBox Noise | 20% | | | 40% | | |
|---|---|---|---|---|---|---|
| Label Noise | 0% | 20% | 40% | 0% | 20% | 40% |
| vanilla (Ren et al., 2015) | 75.5 | 70.7 | 66.9 | 59.3 | 54.2 | 50.0 |
| objectness maximization | 75.8 | 71.1 | 67.4 | 63.2 | 59.6 | 55.7 |
| CA-BBC | **76.8** | **72.4** | **68.0** | **67.8** | **64.7** | **61.7** |

Table 1: Evaluation of class-agnostic bounding box correction. Numbers are mAP@.5 on VOC2007 test set for models trained with different mixtures of label noise and bbox noise.

## 3.4 MODEL TRAINING

Let $Y^*$ and $B^{**}$ denote a mini-batch of soft labels and refined bounding boxes, respectively. We use them as the new GT to train the model. Specifically, we update $\Theta = \{\theta_{\mathrm{cnn}}, \theta_{\mathrm{rpn}}, \theta_d^1, \theta_d^2\}$ to optimize the following losses: (1) the loss function of RPN defined in (Ren et al., 2015), $L_{\mathrm{rpn}}(B^{**})$; (2) the classification loss for the two detection heads, $L_{\mathrm{cls}}^1(Y^*)$ and $L_{\mathrm{cls}}^2(Y^*)$, defined as the cross-entropy loss $\sum_i -y_i^* \log(p_i)$; (3) the localization loss for the two detection heads, $L_{\mathrm{loc}}^1(B^{**}, Y^*)$ and $L_{\mathrm{loc}}^2(B^{**}, Y^*)$, defined as the smooth L1 loss (Girshick, 2015).

## 4 EXPERIMENTS

### 4.1 DATASETS AND IMPLEMENTATION DETAILS

Since most available datasets for object detection have been extensively verified by human annotators and contain little noise, we created noisy annotations using two popular benchmark datasets, PASCAL VOC (Everingham et al., 2010) and MS-COCO (Lin et al., 2014), First, we generated synthetic noise to simulate human mistakes of different severity, by corrupting the training annotation with a mixture of label noise and bounding box noise. For label noise, we follow previous works (Jiang et al., 2018b; Arazo et al., 2019) and generate symmetric label noise. Specifically, we randomly choose $N_l\%$ of the training samples and change each of their labels to another random label. For bounding box noise, we perturb the coordinates of all bounding boxes by a number of pixels uniformly drawn from $[-wN_b\%, +wN_b\%]$ ($w$ is bbox width) for horizontal coordinates or $[-hN_b\%, +hN_b\%]$ ($h$ is bbox height) for vertical coordinates. We experiment with multiple combinations of label noise ranging from 0% to 60% and bounding box noise ranging from 0% to 40%. Under 40% bbox noise, the average IoU between a noisy bbox and its corresponding clean bbox is only **0.45**. For VOC, we use the union set of *trainval2007* and *trainval2012* as training data, and *test2007* as test data. We report mean average precision (mAP@.5) as the evaluation metric. For MS-COCO, we use *train2017* as training data, and report mAP@.5 and mAP@[.5, .95] on *val2017*.

We also mined large amounts of free training data with noisy annotations by using machine-generated annotations on unlabeled images. We first train a Faster R-CNN detector on 10% of labeled data from COCO *train2017*, which has a validation mAP@.5 of 40.5. Then we use the trained detector to annotate *unlabeled2017*, which contains 123k unlabeled images. We use COCO *unlabeled2017* with machine-generated annotations as our noisy training data.

We use the common Faster-RCNN (Ren et al., 2015) architecture with ResNet-50 (He et al., 2016) and FPN (Lin et al., 2017) as the feature extractor. We train the model using SGD with a learning rate of 0.02, a momentum of 0.9, and a weight decay of $1e-4$. The hyper-parameters are set as $\lambda = 0.1$, $T = 0.4$, $\rho = 0.5$, and $\alpha \in \{0, 100, 200\}$, which are determined by the validation performance on 10% of training data with clean annotations (only used for validation). We implement our framework based on the mmdetection toolbox (Chen et al., 2019a). In terms of computation time, our method increases training time by $\sim 24\%$ compared to vanilla training. During inference, we only use the first detection head unless otherwise specified, which does not increase inference time.

### 4.2 EVALUATION ON CA-BBC

First, we evaluate the effect of the proposed CA-BBC method by itself. We train a detector following the proposed learning framework, except that we only perform the first step of noise correction (*i.e.* CA-BBC). Table 1 shows the results on VOC with different mixtures of label noise and bounding box noise. Compared to vanilla training without any noise correction (Ren et al., 2015; Chen et al., 2019a), performing CA-BBC can significantly improve performance, especially

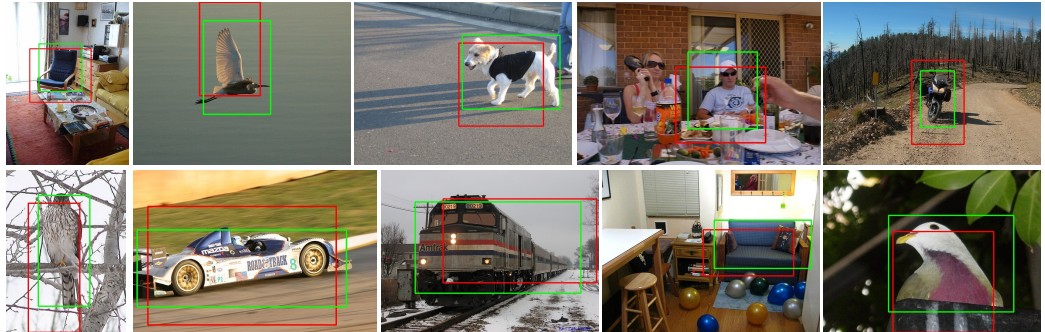

Figure 3: Examples of class-agnostic bounding box correction on VOC with 40% label noise and 40% bounding box noise. Noisy GT bounding boxes are in red and the corrected bounding boxes are in green.

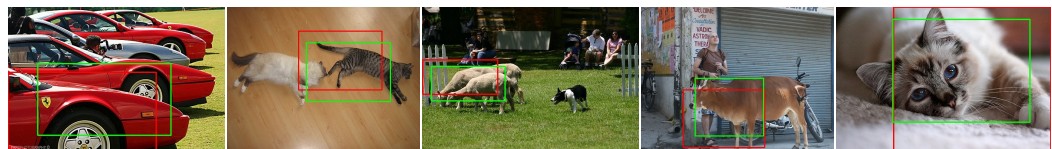

Figure 4: Examples of failed bbox corrections from CA-BBC. Noisy GT bounding boxes are in red and the corrected bounding boxes are in green. CA-BBC could be confused when the GT bboxes cover multiple object instances of the same class. It also sometimes fails to extend the box boundaries to cover the entire object.

| BBox Noise | 0% | | | | 20% | | | | 40% | | | |
|---|---|---|---|---|---|---|---|---|---|---|---|---|
| Label Noise | 0% | 20% | 40% | 60% | 0% | 20% | 40% | 60% | 0% | 20% | 40% | 60% |
| Vanilla (Ren et al., 2015) | 78.2 | 72.9 | 69.3 | 62.1 | 75.5 | 70.7 | 66.9 | 61.2 | 59.3 | 54.2 | 50.0 | 45.9 |
| Co-teaching (Han et al., 2018) | 78.3 | 76.5 | 74.1 | 69.9 | 75.6 | 73.2 | 69.7 | 65.1 | 60.6 | 59.7 | 55.8 | 50.4 |
| SD-LocNet (Zhang et al., 2019) | 78.0 | 75.3 | 73.0 | 66.2 | 75.3 | 72.1 | 67.5 | 64.0 | 59.7 | 58.7 | 54.5 | 49.2 |
| NOTE-RCNN (Gao et al., 2019) | 78.6 | 76.7 | 74.9 | 69.9 | 76.0 | 73.7 | 70.1 | 65.8 | 63.4 | 61.5 | 57.8 | 53.7 |
| Ours | **80.1** | **79.1** | **77.7** | **74.1** | **77.9** | **76.7** | **74.8** | **71.9** | **71.9** | **70.6** | **69.1** | **64.5** |

Table 2: Results on VOC07 test set with different levels of training noise. Numbers indicate mAP@.5.

for higher level of bbox noise. The improvement is consistent despite the increase of label noise, which demonstrates the ability of CA-BBC to disentangle the two types of noise and effectively correct bbox noise. We also demonstrate the effect of the proposed discrepancy minimization by removing $\mathcal{D}(p_1, p_2)$ from the loss in Eq. 2, and only maximize the objectness of the bbox region, which leads to lower performance. Figure 3 show qualitative examples of CA-BBC. The noisy GT bboxes are shown in red whereas the corrected bboxes are shown in green. CA-BBC can update the bounding boxes to more accurately capture the objects of interest.

## 4.3 COMPARISON WITH THE STATE-OF-THE-ART

We evaluate our full learning framework with two-step noise correction and compare it with multiple existing methods for learning with noisy annotations. We implement all methods using the same network architecture. Since previous methods operate in different settings as ours, we adapt them for our problem to construct strong baselines as described in the following:

- Co-teaching (Han et al., 2018) simultaneously trains two models where each model acts as a teacher for the other by selecting its small-loss samples as clean data to train the other. It has been employed by Chadwick & Newman (2019) for training object detectors with noisy data. We adapt co-teaching into our dual-head network, where each detection head selects box samples with small classification loss to train the other head. Note that the RPN is trained on all boxes.

- SD-LocNet (Zhang et al., 2019) proposes an adaptive sampling method that assigns a reliable weight to each box sample. Higher weights are assigned to samples with higher classification scores and lower prediction variance over consecutive training epochs.

| Method | $N_b = 20\%, N_l = 20\%$ | | $N_b = 40\%, N_l = 40\%$ | | machine-generated | |
|---|---|---|---|---|---|---|
| | mAP@.5 | mAP@[.5, .95] | mAP@.5 | mAP@[.5, .95] | mAP@.5 | mAP@[.5, .95] |
| Vanilla (Ren et al., 2015) | 47.9 | 23.9 | 29.7 | 10.3 | 41.5 | 21.5 |
| Co-teaching (Han et al., 2018) | 49.7 | 24.6 | 35.9 | 14.6 | 41.4 | 21.5 |
| SD-LocNet (Zhang et al., 2019) | 49.3 | 24.5 | 35.1 | 13.9 | 42.8 | 21.9 |
| NOTE-RCNN (Gao et al., 2019) | 50.4 | 25.1 | 38.5 | 15.2 | 43.1 | 22.0 |
| Ours | **53.5** | **27.7** | **47.4** | **21.2** | **46.5** | **23.2** |
| Oracle (clean) | 54.5 | 31.4 | 54.5 | 31.4 | 54.5 | 31.4 |

Table 3: Results on COCO val2017 with different levels of training noise.

| Forward Corr. | Dual Heads | CA BBC | Dual Infer. | 0% | | | 20% | | | 40% | | | $(N_b)$ |
|---|---|---|---|---|---|---|---|---|---|---|---|---|---|
| | | | | 20% | 40% | 60% | 20% | 40% | 60% | 20% | 40% | 60% | $(N_l)$ |
| ✓ | | | | 78.9 | 77.4 | 73.4 | 75.9 | 73.6 | 68.8 | 67.2 | 65.3 | 59.1 | |
| ✓ | ✓ | | | 79.1 | 77.7 | 74.1 | 76.2 | 74.1 | 69.8 | 67.9 | 66.0 | 60.3 | |
| ✓ | ✓ | ✓ | | 79.1 | 77.7 | 74.1 | 76.7 | 74.8 | 71.9 | 70.6 | 69.1 | 64.5 | |
| ✓ | ✓ | ✓ | ✓ | **79.6** | **78.3** | **74.8** | **77.3** | **75.2** | **72.5** | **71.3** | **69.8** | **65.4** | |

Table 4: Ablation study to examine the effect of each component in the proposed framework. Numbers indicate mAP@.5 on VOC 2007 test set. The results validate the efficacy of the proposed CA-BBC and dual-head noise correction method. Ensemble of the two detection heads during inference can further boost performance.

- NOTE-RCNN (Gao et al., 2019) uses clean seed box annotations to train the bbox regression head. It also pretrains a teacher detector on the clean annotations for knowledge distillation. Because we do not have clean annotations, we follow previous works (Han et al., 2018; Arazo et al., 2019) and consider box samples with smaller classification loss as clean ones. We first train a detector in a standard manner to mine clean samples. Then we utilize the clean samples following NOTE-RCNN (Gao et al., 2019).

Table 2 shows the comparison results on VOC, where the training data contains different mixtures of label noise and bbox noise. Our method significantly outperforms all other methods across all noise settings. For high levels of noise ($N_b = 40\%, N_l \in \{40\%, 60\%\}$), our method achieves $\sim 20\%$ improvement in mAP compared to vanilla training, and $>10\%$ improvement compared to the state-of-the-art NOTE-RCNN (Gao et al., 2019)

On clean training data with $0\%$ annotation noise, our method can still improve upon vanilla training by $+1.9\%$, mostly due to the proposed soft labels. Compared to the one-hot GT labels, soft labels contain more information about an image region in cases where multiple objects co-exists in the same bounding box. Moreover, using soft labels has the effect of label smoothing, which could prevent overfitting and improve a model's generalization performance (Müller et al., 2019).

Table 3 shows the results on COCO. Our method outperforms all baselines by a large margin. Under $40\%$ of label and bbox noise, vanilla training results in a catastrophic degradation of $-24.8\%$ in mAP@.5 compared to training on clean data (oracle), whereas our method can reduce the performance drop to $-7.1\%$. The proposed method also achieves improvement under machine-generated noise, which validates its practical usage to train detectors by utilizing free unlabeled data.

## 4.4 ABLATION STUDY

In Table 4, we add or drop different components in our framework to examine their effects. Below we explain the results in detail. More ablation study and qualitative results are shown in the appendix.

- In the first row, we perform noise correction with only one detection head, by using its output to create soft labels and regress bounding boxes. Compared with the proposed dual-head network where knowledge is distilled from the ensemble, using a single head suffers from confirmation bias where the model's prediction error would accumulate and thus degrade the performance.
- In the second row, we remove CA-BBC from the proposed framework and only perform the dual-head noise correction (Section 3.3). Compared with the results using the proposed two-step noise correction (the third row), the performance decreases considerably for higher level (40%) of bounding box noise, which validates the importance of the proposed CA-BBC.
- The third row shows the results using the proposed method.

- In the last row, we use the ensemble of both detection heads during inference by averaging their outputs, which leads to further performance improvement.

## 5 CONCLUSION

To conclude, this paper addresses a new challenging research problem, which aims to train object detectors from noisy annotations that contain entangled label noise and bounding box noise. We propose a noise-resistant learning framework which jointly optimizes noisy annotations and model parameters. A two-step noise correction method is proposed, where the first step performs class-agnostic bbox correction to disentangle bbox noise and label noise, and the second step performs dual-head noise correction by self-distillation. Experiments on both synthetic noise and machine-generated noise validate the efficacy of the proposed framework. We believe that our work is one step forward towards alleviating human from the tedious annotation effort.

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
