# OpenReview forum: "Towards Noise-resistant Object Detection with Noisy Annotations"
_ICLR.cc/2021/Conference — Reject_

### Official Review · AnonReviewer1 · 2020-10-27
**This paper addresses the problem of training object detectors from noisy annotations that contains entangled label noise and bounding box noise. The authors propose a two-step noise correction method where the first step tried to correct bounding boxes and the second step performs both label and box correction by self-distillation. The experiments on synthetic and machine-generated noisy datasets show promising results.**

**Rating:** 5
**Confidence:** 5

**Review:**

Pros
- The writing of the paper is good and clear. The problem paper it tries to solve is very practical and meaningful — annotation noise for training object detectors.
- The proposed idea is interesting and the dual heads and two-step training strategy are critical for achieving the good performance. The method is described well with details and references. The paper also provide some intuitive explanation on why the method works.

Cons
- The noisy annotation problem exists in practice and the paper’s motivation is also valid. However the experiments are carried on simulated noisy datasets. Although the results are promising, the practical value is limited. The way it simulates the noise may not reflect the most common case in practice. Since most object detection datasets are usually annotated in two steps where the first step identifies the class labels for instances in each image and the second step tries to draw bounding boxes for each present category. Even if the label noise happens in the first step, the second step is able to alleviate it by skipping drawing bounding box for that category. Thus the bounding box noise might be more common in real datasets and missing annotation is also another source of noise we should consider.
- Different classes may suffer differently to the noise. Objects of different sizes also suffer differently. The paper lacks the analysis on class-specific performance and object size-specific performance on different noise settings.
- In Table 2, when training on clean dataset, the proposed method still outperforms the vanilla training, which is somewhat surprising. It would be interesting to investigate the real reason behind this result by removing different components from the proposed method.
- Since one-stage detectors also play an important role in real applications due to their efficiency. How will this method perform on one-stage detectors?
- It would be interesting to see more qualitative results to compare the proposed method and the state-of-the-art methods.

---

> ### Author Response · Authors · 2020-11-17
> **Response to Reviewer #1**
>
> We appreciate the reviewer for the positive comments and valuable feedback. Here we address the concerns one by one.
>
> - Firstly, we agree with the reviewer that two-step annotation can alleviate label noise to some extent. However, our method aims to improve annotation efficiency in practice, which can be achieved by (1) single-step annotation, (2) using weakly-labeled Web images, (3) using machine-generated annotations. All of these methods would yield label noise.
> Secondly, we also evaluated our method on machine-generated noise, where we achieve state-of-the-art performance.
>
> - We thank the reviewer's suggestion and provide the object size-specific performance as follows. The size definition follows the COCO standard. We observe that objects of different sizes suffer similarly to the noise. We also observe that different classes suffer similarly to the noise, and do not show the detailed results here due to the  large number of classes.
> | noise ratio | overall mAP| small mAP | medium mAP | large mAP|
> |----------------|:---------------:|:---------------:|:---------------:|:---------------:|
> |40%|21.2 |12.4|24.5|26.7|
> |20%|27.7 | 16.3|31.1|35.9|
> |0%|34.9|19.8 | 38.5 |45.4|
>
> - Our proposed method outperforms the vanilla training on clean datasets. The improvement in mAP is 1.9% on VOC and 3.5% on COCO. The improvement can be mostly attributed to using the model’s prediction to generate soft labels. Training on the soft label is a form of self-distillation. The soft label contains rich information about (1) secondary objects that possibly co-exist in the same bounding box as the main object, and (2) semantic similarity information between different classes. Furthermore, it has the effect of label smoothing, which has been shown to result in a well-calibrated model that generalizes better.
>
> - The proposed training method requires a differentiable ROI-pooling operation, hence cannot be directly applied to one-stage detectors. We hypothesize that one-stage detectors are more prone to noise because they lack the flexibility to adjust bounding box predictions, and hence focus our study on two-stage detectors.
>
> Thanks again for your review. Please let us know if we have addressed your concerns or if you have other questions.

---

### Official Review · AnonReviewer2 · 2020-10-28
**This work presents an object detection framework which is well suited for noisy environment when we have class label noise as well as bounding box noise. The results are motivating, but the approach lacks technical novelty with some assumptions which are not well motivated.**

**Rating:** 5
**Confidence:** 4

**Review:**

In this work the authors propose a framework to perform object detection when there is noise present in class labels as well as bounding box annotations. The authors propose a two-step process, where in the first step the bounding boxes are corrected in class-agnostic way, and in the second step knowledge distillation has been used to correct the class labels. The propose method has been evaluated on two different datasets with synthetic noise.

Pros:

The idea of class-agnostic noise correction for bounding-box is interesting as it avoids the effect of noisy class label.

The idea of knowledge distillation has been extended for correcting bounding boxes which has some novelty.

The evaluation is comprehensive with ablations for all the components of the proposed framework.

Cons:

The class-agnostic update for bounding box is based on the assumption that, if a bounding box tightly covers an object, then two diverged classifiers would agree with each other and produce the same prediction. It is not clear why this assumption should hold true. Also the assumption that both classifiers would have low scores for the background class,i.e., high objectness scores seems difficult to understand as if the classifiers are trained on noisy labels, how can we be confidence in their predictions?

The knowledge distillation for noisy learning has been well studied and is not a significant novel contribution. The extension to bounding box correction is new but it does not include any novel ideas.

In general, this is an interesting work and solves a very interesting problem. But some of the assumptions used in the class-agnostic bounding-box correction are not easy to understand and also not well motivated. The results are motivating, but as such there is not enough novelty in the proposed solution which stands out.

-- post rebuttal---
After carefully reading the authors response, I still think the assumptions are not well motivated. For example, 'On a clean bounding box, even a less well-trained classifier can produce a lower-entropy (high-confidence) prediction with low background scores. Furthermore, it is less likely that two different classifiers both predict the same wrong class with high scores.'. Both of these assumptions seems very strong. They might hold sometimes, but they can not be generalized. Also, I still think the technical novelty is not enough in this submission. Therefore, I am not very positive about this submission.

---

> ### Author Response · Authors · 2020-11-17
> **Response to Reviewer #2**
>
> We appreciate the reviewer for the positive comments and valuable feedback. Here we address the concerns.
>
> Firstly, our experiments in Table 1 verify our assumptions that both (1) minimizing classifier discrepancy and (2) maximizing objectness score are effective in correcting the noisy bounding boxes. Secondly, we experimentally observed that the classifier tends to produce a high-entropy (low-confidence) prediction when a bounding box is inaccurate. On a clean bounding box, even a less well-trained classifier can produce a lower-entropy (high-confidence) prediction with low background scores. Furthermore, it is less likely that two different classifiers both predict the same wrong class with high scores. These observations motivated us to design a bounding box correction algorithm which encourages high-confidence and consistent predictions between two classifiers.
>
> In terms of the technical novelty, our class-agnostic bounding box correction is a novel method. We use the gradient from the classifiers’ objective to directly update the bounding boxes, which has not been proposed in existing works. We agree with the reviewer that our self-distillation method is similar to some of the existing methods in label noise learning, which we have discussed in the paper. However, we are the first to adapt the self-distillation idea to object detection, and demonstrate strong empirical performance. Even on clean datasets, our self-distillation method can achieve substantial improvements compared to vanilla training, with an mAP improvement of 1.9% on VOC and 3.5% on COCO.
>
> Thanks again for your review. Please let us know if we have addressed your concerns or if you have other questions.

---

### Official Review · AnonReviewer4 · 2020-10-30
**Solid approach but with gaps in the experiments**

**Rating:** 6
**Confidence:** 4

**Review:**

This paper addresses the task of training object detectors from noisy labels. Unlike other methods that operate in the weakly- or semi-supervised regime, this method operates on the full set of bounding box annotations but assumes that all of them have been synthetically corrupted:
- Each bounding box coordinate is drawn from a uniform distribution centered around the ground truth coordinate and with a range that is some pre-specified fraction of the bounding box height/width.
- Each class label is either retained or flipped to some random class, with the decision to flip drawn from a Bernoulli distribution.

To nonetheless reliably train an object detector from this data, the following procedure is proposed:
- preparation: A Faster R-CNN detector with a fully differentiable feature pooling layer (PrRoi) is extended with a second detection head. This detector is trained for a warm-up period on the corrupted labels in the usual fashion.
- for each mini-batch:
  - two-stage noise correction:
    1. class-agnostic bounding box correction: Both detection heads classify the incoming bounding box proposal from the RPN. If both agree that it is background, the proposal is discarded. Otherwise, with the help of the differentiable pooling layer the bounding box coordinates are optimised to maximise agreement of softmax scores between both detection heads. The optimisation runs for a single step to save time.
    2. label update and second bounding box update: given the corrected bbox from the last step, the class predictions of both heads are averaged and "sharpened" to reduce entropy, and the regressed bounding boxes are averaged together with the ground truth box
  - updating the network parameters:
    - Given the corrected labels and (twice-corrected) bounding boxes, the network parameters are updated in the usual manner.

This approach is evaluated against other weakly-supervised object detection approaches and performs quite favourably across multiple noise levels. As far as I can tell, the authors re-implemented the baselines for a fairer comparison, which is a plus. So overall, the results are quite good and the approach is sensible and seems to work. I don't have any major deal-breaking complaints, but a series of smaller ones mainly regarding gaps in the experiments:

The introduction motivates this approach by citing a number from a 2012 paper that is very much outdated and should be replaced. According to Su et al., the median time for a single bounding box annotation is 42 seconds. In the much more recent OpenImagesV4 paper (Kuznetsova et al., 2018), they find that they need 7s on average per bounding box with (among other things) a much more efficient box drawing procedure, which is well below the average of 88s from Su et al. 2012. This more recent number also tracks with my own annotation experience.

Given this emphasis on wanting to save annotation time, I would have expected the "machine-generated" annotations to figure much more prominently in the experiments (i.e. train a network on 10% of the ground truth and use detections on the rest of the training images). These only appear once in the sota comparison. It is still interesting to see that the proposed method handles high synthetic noise significantly better than competing methods, but this is the less realistic setting as the synthetic noise is applied to the entire set of ground truth annotations. I would thus also dispute the description of this setting as being "more challenging and practical" than other settings considered for weakly-supervised training (e.g. mostly relying on image-level labels).

The warm-up phase seems like a critical part of this approach and it's only mentioned in passing without any discussion. How much does the length of the warm-up phase matter? Did you experiment with this? Is there a trade-off between (a) getting some amount of training for the correction to work, vs. (b) over-training on the noisy labels? Is the recommended warm-up length dependent on the amount of noise?

The first noise correction stage (CA-BBC) is explicitly based on the assumption that if a bounding box tightly covers an object, the two classifiers will agree. This is a testable assumption and I would have been curious to see to what degree it actually holds.

Given that the proposed method focuses on correcting the bounding box annotations, I think it would have been important to report results on the corrected training boxes, especially since you have access to the un-corrupted annotations. Based on the final results, the method obviously must be doing something right but some distribution of IoU values before and after the correction/training process would have been interesting w/ some more qualitative examples of boxes that weren't successfully corrected.

I may have overlooked this, but since training involves multiple epochs (and with it multiple corrections to the same bounding box) are the corrections retained or discarded? That is: Are individual boxes cumulatively corrected throughout the training?

Alpha is the hyperparameter for the step size for CA-BBC and in section 4.1 you specify three different values for it {0, 100, 200}. Why three values?

The first result table is inconsistent with others, only reporting a subset of the results. What happens in the 0% label noise case?

The optimisation in CA-BBC is only run for one step for efficiency reasons. What about the effect on performance? Do additional steps bring any improvements? Currently, the noise correction only adds 25% computation time to training so I assume that running the optimisation for a few more steps won't make the approach prohibitively slow.

Corrections/editing suggestions:
- A correction needs to be made to the Chadwick & Neuman reference, as the paper was published at IEEE IV 2019.
- p2, par 2: "distil" -> "distill"
- 4.3 title: "state-of-the-arts" -> "state-of-the-art"
- I would maybe re-organise the experimental section a little, as 4.2 focuses on the CA-BBC part on its own, the 4.3 compares the full method against the state-of-the-art, and 4.4 returns to a more complete ablation study. I would switch the 4.3 and 4.4, i.e. first get all the ablations out of the way then compare against the state-of-the-art.
- Fig. 1 shows one part of the method in detail and Fig. 2 an overview of full method. I would either switch the order here, or merge these somehow.

Conclusion:
-------------------

Overall, I think this is a sensible aproach to training with noisy bounding box annotations, which compares favourably against competing methods. I have several reservations with regards to the experiments, especially the emphasis on synthetic noise as opposed to the more realistic setting where only a small set of hand-annotated labels are available. I would have liked to see more analysis (esp. quantitative) of the corrections of the training data, as this is what the method does. The warm-up phase appears to be critical and is also given short shrift. These are all things that are not fundamental problems, hence the (slightly) positive rating.

---

> ### Author Response · Authors · 2020-11-17
> **Response to Reviewer #4**
>
> We appreciate the reviewer’s positive comments and insightful feedbacks. Here are our responses to the concerns:
>
> > 1. The OpenImagesV4 paper (Kuznetsova et al., 2018) found that they need 7s annotation time on average per bounding box.
>
> We thank the reviewer for this suggestion and have updated our paper accordingly.
>
> > 2. How much does the length of the warm-up phase matter? Is there a trade-off between (a) getting some amount of training for the correction to work, vs. (b) over-training on the noisy labels? Is the recommended warm-up length dependent on the amount of noise?
>
> In general, we find our method not sensitive to the length of the warm-up. For different noise ratios, we use a constant warm-up period of 5000 iterations for VOC and 10000 iterations on COCO. We find that extending the warm-up period by a few thousand iterations does not affect the overall result. However, performing noise-correction too early does hurt performance as the model has not learned enough useful knowledge.
>
> >3. The first noise correction stage (CA-BBC) assumes that if a bounding box tightly covers an object, the two classifiers will agree. I would have been curious to see to what degree it actually holds.
>
> Firstly, our experiments in Table 1 demonstrate that noisy bounding boxes can be corrected by minimizing the discrepancy between two classifiers. Secondly, we have experimentally observed that on datasets with less bounding box noise, the two classifiers tend to produce higher-confidence predictions with higher agreement.
>
> >4. It would have been important to report results on the corrected training boxes.  Some distribution of IoU values before and after the correction/training process would have been interesting w/ some more qualitative examples of boxes that weren't successfully corrected.
>
> Following the valuable suggestion, we have measured the IoU between the noisy bounding boxes and the clean bounding boxes before and after the training process. Under 40% bounding box noise on VOC, the average IoU before training is only 0.45. After training, the corrected bounding boxes have a much higher IoU of 0.76 with clean bounding boxes.
> Following the reviewer’s suggestion, we added more qualitative examples in Figure 4 where the proposed CA-BBC failed to correct the bounding box noise. Due to the optimization objective (i.e. maximize classifier agreement and objectness score), It could be confused when the GT bounding box covers multiple object instances of the same class. It also sometimes fails to extend the box boundaries to cover the entire object.
>
>
> >5. Are the corrections retained or discarded? That is: Are individual boxes cumulatively corrected throughout the training?
>
> We used the current-epoch correction for each individual box and discarded previous ones. We have also experimented with using an exponential-moving-average to ensemble predictions temporally, but found that to have minimal improvements.
>
> >6. Why three values for alpha?
> Response: Alpha controls the step size for bounding box correction, thus its value needs to be adjusted based on the level of bounding box noise. In our experiments, we set alpha=0 for 0% bbox noise, alpha=100 for 20% bbox noise, and alpha=200 for 40% bbox noise.
>
> For Table 1, what happens in the 0% label noise (authors: I believe the reviewer meant bbox noise) case?
> Response: Table 1 shows the results where only the 1st stage CA-BBC is performed. With 0% bbox noise, CA-BBC will not have any effect hence we did not report its results.
>
> >7. Do additional CA-BBC steps bring any improvements?
>
> Yes, some additional gain can be achieved if we perform 2 steps of CA-BBC update. However, the gain is relatively marginal hence we used a single step for computational efficiency.
>
>
> 8. We appreciate the very helpful editing suggestions and have modified our paper accordingly.
>
> Thanks again for your review. Please let us know if we have addressed your concerns or if you have other questions.

---

### Decision · Program_Chairs · 2021-01-07
**Final Decision**

**Decision:**

Reject

**Comment:**

The paper received borderline scores, before and after the rebuttal. Thus, support for paper acceptance isn't sufficiently strong. While the reviewers see merit, concerns which remain after the discussion phase, include how convincing the experimental settings and results are, and uncertainty about the motivation and practical value.